# Elimination of Pathogen Biofilms via Postbiotics from Lactic Acid Bacteria: A Promising Method in Food and Biomedicine

**DOI:** 10.3390/microorganisms12040704

**Published:** 2024-03-30

**Authors:** Jiahao Che, Jingjing Shi, Chenguang Fang, Xiaoqun Zeng, Zhen Wu, Qiwei Du, Maolin Tu, Daodong Pan

**Affiliations:** 1State Key Laboratory for Managing Biotic and Chemical Threats to the Quality and Safety of Agro-Products, Ningbo University, Ningbo 315832, China; c1045982171@163.com (J.C.); sjjing812@163.com (J.S.);; 2Key Laboratory of Animal Protein Food Processing Technology of Zhejiang Province, College of Food Science and Engineering, Ningbo University, Ningbo 315832, China; fcgfcghhh20030830@163.com; 3Zhejiang-Malaysia Joint Research Laboratory for Agricultural Product Processing and Nutrition, Ningbo University, Ningbo 315832, China

**Keywords:** postbiotics, bioactive substances, antibiofilm agents, food safety, biomedical function

## Abstract

Pathogenic biofilms provide a naturally favorable barrier for microbial growth and are closely related to the virulence of pathogens. Postbiotics from lactic acid bacteria (LAB) are secondary metabolites and cellular components obtained by inactivation of fermentation broth; they have a certain inhibitory effect on all stages of pathogen biofilms. Postbiotics from LAB have drawn attention because of their high stability, safety dose parameters, and long storage period, which give them a broad application prospect in the fields of food and medicine. The mechanisms of eliminating pathogen biofilms via postbiotics from LAB mainly affect the surface adhesion, self-aggregation, virulence, and QS of pathogens influencing interspecific and intraspecific communication. However, there are some factors (preparation process and lack of target) which can limit the antibiofilm impact of postbiotics. Therefore, by using a delivery carrier and optimizing process parameters, the effect of interfering factors can be eliminated. This review summarizes the concept and characteristics of postbiotics from LAB, focusing on their preparation technology and antibiofilm effect, and the applications and limitations of postbiotics in food processing and clinical treatment are also discussed.

## 1. Introduction

The virulence and drug resistance of pathogens are closely related to their ability to form biofilms [1,2], which is a major challenge for the food and medical industries [3,4]. A biofilm is a complex biological community composed of single or distinct microorganisms [5] that adsorb onto the surface of host cavities or inert or active materials, and secrete an extracellular matrix. Microbial cells are wrapped in the biofilm [6,7] to enhance their capacity to adapt to the environment [8]. The complex process of biofilm formation is controlled by the quorum sensing (QS) intercellular communication system [9,10], which comprises complex signaling networks involving multiple secondary messengers [11], such as cyclic (c)-di-GMP and cAMP [12]. A biofilm comprises microbial cells and self-secreted extracellular matrix (ECM), which is primarily composed of extracellular polysaccharides (EPSs), extracellular DNA (eDNA), secreted proteins, and lipids [13]. A biofilm can not only shield pathogens from an unfavorable environment [14], but also increases their drug resistance, and acts as a natural barrier to bacterial growth [15].

Raw and auxiliary material feeding pipes and worktable surfaces are vulnerable to biofilm attachment by a variety of foodborne pathogens during food production. In the protective environment provided by a biofilm [16], it is typically hard to kill these foodborne pathogens using conventional antibiotics [17], which poses a threat to product quality and safety [18]. Bacterial or fungal biofilms are also widely known to contribute to chronic human infections and diseases [19], such as lung and urinary tract infection [20,21] and skin and gastrointestinal inflammation [22], which cannot be effectively treated using conventional antibiotics. Biofilms are easily formed on the surfaces of implantable medical devices [23] and human tissues and organs, which are potential sources of infection in the clinical environment. Dental caries is caused by biofilms attached to the enamel surface and has been one of the most serious global public health issues [24] because of its high prevalence and potential impact on general health. Consequently, to prevent the adhesion of pathogens to food contact surfaces and the formation of biofilms in the clinical environment, new control methods are urgently required.

Since the concept of postbiotics was first proposed, the topic of postbiotics from lactic acid bacteria (LAB) has been at the forefront of research. Traditional antibiotics and living antimicrobial agents have limited control effects on biofilm infection, and are associated with several problems, such as the drug resistance of pathogens and the inactivation of probiotics. Meanwhile, new antimicrobial agents based on biosurfactants (BSs) derived from LAB have attracted wide attention. Metabolites and the cellular composition of LAB, defined as postbiotics, have been discovered to hinder the biofilm formation of pathogens by disrupting cellular communication and regulating the process of ECM formation. The complexity of inhibiting early biofilm formation by postbiotics from LAB is attributed to the abundance of metabolites involved, and it is more direct and effective than using living LAB. Therefore, postbiotics based on the metabolites of LAB could be widely used in the removal of biofilms in the fields of food and clinical medicine [25] because of their good absorption, high safety, and stability [26]. There are promising future applications for these probiotic-derived substances. From the perspective of the biofilm cycle of pathogens, this review provides an overview of postbiotics, focusing on their preparation and antibiofilm mechanism. It also discusses their potential applications in the fields of food and medicine. Finally, a series of problems existing in the current functional postbiotics are pointed out, aiming to provide direction for the development of postbiotics in the future.

## 2. The Formation Mechanism of Biofilm

### 2.1. High Correlation with Quorum Sensing

The formation of a bacterial biofilm customarily occurs in four stages, and the QS system controls the last three stages. Fimbriae and adhesins are usually responsible for the initial attachment of bacteria, while the production of dextran and eDNA promotes the maturation of the biofilm. Figure 1 shows the bacterial biofilm formation process and its control methods of each stage.

With the increase in cell density and a change in the growth cycle, bacteria and fungi secrete one or more chemical signal molecules that are used for intraspecific or interspecific communication to coordinate group behavior, recognized as QS [27]. Specific QS signals can activate a variety of physiological and biochemical reactions, including biofilms, secondary metabolites, ECM production, and the QS system itself, which is an essential link in the virulence process [28].

Bacterial QS systems can be categorized into three common groups. The LuxI/R signal system uses N-acyl-homoserine lactones (AHLs) as signal molecules. Small molecular peptides are more prevalent in Gram-positive bacteria, such as LAB, as signal molecules of the auto-inducing peptide (AIP) signaling system. The LuxS/AI-2 signaling system is a universal QS system of autoinducer-2 (AI-2) bacteria used for interspecific communication. Autoinducers enable bacteria to perceive and respond to environments continuously, and to coordinate colony behavior by modifying gene expression [29]. Quorum sensing molecules (QSMs) of fungi control the transformation of the germ tube, mycelial phase, and yeast state, as well as biofilm formation. In Candida albicans, a representative of the pathogenic fungi, farnesol and tyrosol are QSMs that have antagonistic effects. During the biofilm formation of *C. albicans*, the morphological transformation of cells is very important. Consequently, QSMs of *C. albicans* are closely related to biofilm formation.

### 2.2. Initial Adhesion

Bacteria typically adhere to surfaces in two ways. The first is binding to host surfaces via outer membrane adhesion proteins, and the second is adhesion of EPS to host surfaces, such as polysaccharide intercellular adhesion. Adenosine triphosphate (ATP)-stimulated cell lysis and extracellular DNA (eDNA) release play a crucial role in bacterial adhesion and biofilm formation [30].

The initial adhesion of bacteria can be further divided into two stages: the reversible attachment stage, in which bacteria attach non-specifically to the surface; and the irreversible attachment phase, in which bacterial adhesins, such as mucin-binding proteins encoded by a lineal homologous protein coding gene cluster, are utilized. The binding of bacterial surface molecules to the mucus layer is one potential mechanism of adhesion in the host [31], e.g., fimbriae and lipopolysaccharide interact between cells and the surface.

### 2.3. Formation and Maturation of Microcolonies

ECM is responsible for cell adhesion and the formation of the cell scaffold biofilm, thereby maintaining the biofilm’s three-dimensional structure. The expression of ECM secretion-related genes is regulated by QS [32]. The aggregation ability of bacteria, which includes the auto-aggregation ability of the same strain and the co-aggregation ability of different strains, is the basis for the formation of a biofilm.

A cross-kingdom biofilm, such as bacterial–fungal co-aggregation, increased the cariogenic potential of *Streptococcus mutans* [33]. *S. mutans* and *C. albicans* could form mixed-species biofilms, Fungi secrete polysaccharides to promote *S. mutans* colonization in the host [34]. Farnesol, a fungal signal, stimulates the formation of *S. mutans* microcolonies by inducing an increase in the levels of the secondary messenger c-di-GMP, leading to an increase in dextran production and biofilm formation [35,36]. The supernatant of *Lactiplantibacillus plantarum* affects glucose metabolism and inhibits the expression of the virulence gene of *C. albicans*, which encodes an agglutinin-like protein [37]. Maan et al. [38] discovered that EPS provides dual adaptive advantages for cells to form biofilms, which not only promote the co-aggregation of related species, but also inhibit the growth of incompatible species.

Z-form extracellular DNA (Z-DNA) is a structural component of the bacterial biofilm matrix [39]. Z-DNA accumulates with the maturation of the biofilm and, by stabilizing the DNABII protein, contributes to the structural integrity of the biofilm matrix. Different pathogenic bacteria form biofilms by anchored co-aggregation, and their coexistence is ordinarily associated with disease pathogenesis [40]. The co-culture of specific *Lactobacillus* spp. and pathogens with high auto-aggregation ability can significantly inhibit their ability to form biofilms, which might be related to interfering with intraspecific signal transmission and interspecific signal synthesis [41]. The impact of secondary metabolites from *Lactobacillus* spp. on pathogen biofilms was found to be insignificant in the mature stage.

### 2.4. Biofilm Dispersion

In this stage, the microbial cells within the biofilm proliferate and disperse rapidly, ultimately leaving the biofilm in the form of planktonic cells, which is conducive to the transfer of bacteria to new areas and the spread of infection [42]. When nutrients are scarce or the population density is excessive, biofilm bacteria use a dispersal mechanism to separate a portion of the biofilm [43]. The contagious nature of pathogenic biofilms has led to their recognition as a crucial element in the dissemination of disease, both in terms of their formation and dispersion. The study utilized imaging screening techniques to confirm the presence of three distinct dispersed components in undiffused *Vibrio cholerae* mutants, namely signal transduction proteins, matrix degradation enzymes, and movement factors [44]. The signal produced by the matrix initiates the process of matrix digestion, leading to cellular motility and subsequent escape from the biofilm.

## 3. The Summary of Postbiotics from LAB

### 3.1. The Definition and Characteristics of Postbiotics

The term “postbiotics” pertains to the bacterial probiotics and/or cellular components that have undergone biotechnological treatment resulting in inactivation, with or without the inclusion of their metabolites [45]. It remains uncertain whether distinct bacterial metabolites can be classified as postbiotics. The components of postbiotics are characterized by their intricate nature and abundance of bioactive metabolites, comprising both high- and low-molecular-weight factors. The technology behind postbiotic preparation primarily involves the alteration of bacterial cell components through physical or chemical means, resulting in modifications to their structure and function. This process renders the bacteria incapable of growth and reproduction, while still retaining their original beneficial properties. Additionally, postbiotics have been shown to impede the development of pathogenic biofilms, diminish pathogenic virulence, and enhance the equilibrium of the internal environment.

Figure 2 shows the inactivation mode and postbiotic preparation process of LAB. The extraction of postbiotics of LAB mainly involves two steps: inactivation and concentration. The process of heat inactivation and ultrasonic crushing is commonly employed to disrupt the cell membrane of probiotics, thereby facilitating the release of intracellular substances, enabling the preservation and concentration of active cellular constituents [46]. The high-pressure homogenization with faster extraction time and higher efficiency will be widely used in postbiotic extraction, which can protect the bioactive substances to the maximum extent. At present, the concentration in actual industrial production is mainly low-temperature vacuum concentration, which rapidly evaporates most of the water in the fermentation broth to facilitate the preparation of the follow-up powder. As postbiotics are mainly composed of some small molecular bioactive substances, techniques such as selective interception of a specific molecular weight or separation of nanofiltration and dialysis have also been used as pre-treatment of vacuum concentration in recent years. The column chromatography of polypeptides in postbiotics is currently only applicable to the laboratory environment.

The utilization of postbiotics can result in a probiotic effect [47]. Compared with conventional living probiotics, postbiotics have a range of benefits [48]. First, postbiotics have a clear chemical structure, safe dose ranges, and stable safety control. Second, postbiotics have good distribution of organisms, metabolism, and absorption. Studies have shown that postbiotics are conveniently absorbed by the intestine [49,50]. Most importantly, postbiotics derived from LAB show the capacity to impede the development of pathogenic biofilms and demonstrate a dose–response relationship. The postbiotics derived from LAB have potential as secure antibiofilm agents, capable of eliminating pathogenic biofilms while simultaneously exerting a probiotic effect within the human oral cavity and intestinal tract when administered orally. Postbiotics produce equivalent probiotic effects to living bacteria while circumventing issues such as low bioavailability, unstable efficacy, and the potential for transmission of drug resistance genes.

### 3.2. The Postbiotic Efficacy Produced by LAB

LAB, serving as probiotics, exhibit diverse physiological functions that can enhance intestinal function, suppress the proliferation of pathogenic bacteria, and regulate immunity [51,52]. The secondary metabolites of LAB, particularly *Lactobacillus* spp., possess inhibitory properties against the biofilm formation of various foodborne pathogenic bacteria. Additionally, they have demonstrated efficacy in preventing dental caries [53,54], particularly in the early stages of dental plaque formation. Through a large number of in vitro tests and human clinical verification, Yuanda’s biological research team found that Probio-MT (Yuanda Biotechnology, Qingdao, China) has a significant inhibitory effect on oral pathogenic bacteria *S. mutans*, *Porphyromonas gingivalis*, *Fusobacterium nucleatum*, *Actinobacillus actinomycetemcomitans*, and so on, and can be used in the adjuvant treatment of dental caries and other oral diseases. This suggests a potential application of LAB in the field of oral health. The use of postbiotics derived from LAB presents a promising alternative for managing *Listeria monocytogenes* in the field of food processing. Diverse *Lactobacillus* spp. cell-free supernatants (CFSs) have been shown to considerably diminish the expression of the regulatory factor, the QS system gene cluster. The inhibition of biofilm formation and the disruption of pre-formed biofilm integrity in *L. monocytogenes* was observed upon targeting of related genes in biofilm cells [55]. Nonetheless, the inhibitory mechanism remains poorly explained.

At present, conventional approaches to controlling foodborne pathogens are constrained by certain drawbacks, including the utilization of chemical preservatives and thermal processing, which might have an unfavorable impact on the organoleptic properties of food products [56,57]. The utilization of natural LAB-derived antimicrobial agents has the potential to impede the formation of pathogenic bacteria biofilms on non-biological surfaces that encounter raw materials in the food industry. The bacterial strain *Streptococcus lactis* PA003 was shown to produce lactic acid and antimicrobial peptides, which enable it to prevent the formation of biofilms by various foodborne pathogens on non-living surfaces, such stainless-steel sheets, polyvinyl chloride sheets, and glass slides. This is achieved through mechanisms such as exclusion, replacement, and competition [58].

Significantly, experiments have demonstrated the capacity of *Lactobacillus* spp. metabolites to impede the proliferation of *Candida* spp. and the formation of its biofilm in vitro, as well as to diminish its colonization in vivo [59]. This has promising implications for its potential clinical utility. Research has demonstrated that enterococcal strains exhibit a robust capacity for auto-aggregation, and their CFS can decrease the adhesion of numerous pathogenic bacteria that form biofilms [60]. *Lactobacillus* spp. have potential as postbiotic agents in the field of oral healthcare, particularly in the treatment of periodontal pathogens. The CFS concentration of *Lactobacillus* sp. proportionally reduces the biofilm formation by *P. gingivalis* [61]. PostbioYDFF^®^-3 (Yuanda Biotechnology, Qingdao, China), as a commercial postbiotic from LAB for food antisepsis, it can significantly inhibit the formation of *Yersinia enterocolitica* biofilm, in which the inhibition rate of 10 mg/mL is 76.4%. In conclusion, LAB natural antimicrobial agents have the potential to control the development of bacterial pathogen biofilms. The development of a postbiotic from a specific LAB strain that incorporates diverse natural antibiofilm substances could offer a superior and efficient solution for biofilm abrogation during food production, as suggested previously [62].

## 4. Mechanism of Biofilm Control by LAB Postbiotics

As a research hotspot, bioactive substances contained in the secondary metabolites of LAB are regarded as postbiotics, which have potential application in controlling biofilm formation by pathogens. The safety and stability of metabolites and inactivated cells from LAB have been widely studied. It was reported that the metabolite of *Lactobacillus parasitum* HL32 can kill *P. gingivalis* and heat-inactivated *Lactobacillus* sp. still maintains antiadhesion and antibiofilm properties against oral cariogenic bacteria, exerting a positive impact on the oral microflora without additional risk factors compared with living bacteria. The antimicrobial and antibiofilm activity of the postbiotics in lyophilized CFS remained unaffected by high temperature and acidity, even after prolonged storage [63], rendering them particularly suitable for direct use.

Postbiotics from LAB have been observed to hinder the biofilm formation of pathogens through the presence of multiple key constituents, such as short-chain fatty acids, EPS, bacteriocin, lipid wall phosphate, glycolipids, and glycoproteins [64]. To achieve optimal inhibition of pathogenic bacterial biofilm formation, it is essential to disrupt the QS system and regulate the process of ECM formation [65]. The complexity of inhibiting early biofilm formation by LAB is attributed to the abundance of metabolites involved. The mechanism encompasses various strategies, such as impeding reversible adhesion, curtailing the production of extracellular polymers, and quorum quenching (QQ).

QQ can inhibit the synthesis and receptor binding of signal molecules, or degrade and modify them, thereby interfering with the QS system [66]. Quorum sensing inhibitors (QSIs) are small molecules that inhibit the expression of QS-regulated genes [67]. QQ decreases the relative hydrophobicity of cells and weakens the adhesion of the biofilm to the surface of the carrier [68]. Future research will focus primarily on determining the mechanism of a variety of metabolites generated by specific LAB strains that inhibit the formation of biofilms by pathogens. The antibiofilm mechanism and application of LAB postbiotics in food and clinical treatment are presented in Table 1.

During their growth, LAB secrete a range of small molecular organic acids, bacteriocin, lipid derivatives, and other bioactive substances, as well as universal signal molecule AI-2. From the perspective of cellular communication, these secondary metabolites block signal transduction via the degradation of biofilm-forming pathogen QSM, inhibition of the QSM synthesis pathway, and binding to signal molecular receptors, ultimately causing QQ and downregulation of the expression of genes related to biofilm formation and virulence. At the same time, the action of postbiotics stimulates LAB to form a protective biofilm, hinder the recognition of pathogens and host cells, and strengthen host immunity to control pathogenic biofilms, which consolidate the stability of the internal environment. Moreover, these bioactive compounds exert their effects on the exopolysaccharide secretion of microorganisms and the constituents essential for the development of biofilm architecture, leading to the disruption of nascent biofilms, eradication of planktonic cells, and ultimately, regulation of the biofilm formation of pathogens. Figure 3 shows the mechanism of controlling biofilm-forming pathogens by a variety of bioactive substances in postbiotics from LAB.

### 4.1. Inhibition of the QS Pathway

LAB regulate the biofilm formation of pathogens by interfering with the synthesis of signal molecules. QSMs are generated through the utilization of specific substances as substrates, followed by catalysis by diverse enzymes. Consequently, the synthesis of signal molecules can be blocked by either substrate destruction or enzyme activity inhibition.

Postbiotics from *Lactobacillus* spp. showed potential therapeutic application in the prevention of *C. albicans* infection. A small molecule, 1-ethoxycarbonyl-β-carboline, from *Lactobacillus* sp. could prevent yeast-to-filamentous growth transition in *C. albicans* by inhibiting a DYRK1-family kinase, Yak1 [78]. The formation of *C. albicans* mycelia was significantly inhibited by the cell-free extract of *Lacticaseibacillus rhamnosus* [73]. The impact of probiotic *Lactobacillus* sp. on the differentiation of yeast hyphae and biofilm formation of *C. albicans* was investigated using planktonic cell suspension and CFS. The results indicated that in the early stage, postbiotics from LAB had a detrimental effect on *C. albicans* colonization on the host surface and the virulence related to yeast filamentous growth [79].

Meanwhile, certain enzymes in postbiotics could degrade signal molecules, which subsequently diminished the potency of signaling molecules. This resulted in the inability of the bacterial QS system to detect signaling molecules, preventing the initiation of related gene expression [80]. Research has revealed that four distinct categories of enzymes possess the capacity to break down or neutralize AHL signals. Two enzymes, AHL-lactonase and AHL-decarboxylase, could hydrolyze the lactone ring of signaling molecules [81,82]. Two additional enzymes, AHL-acyltransferase and AHL-deaminase, have the ability to hydrolyze the acyl side chain of signaling molecules [83]. Although numerous QQ enzymes that are capable of degrading AHL have been identified in probiotic bacteria [84], further excavation of LAB is needed.

QQ biological stimuli with conserved parts similar to pathogenic bacteria QSM can stimulate local QQ bacteria to produce corresponding degradation enzymes or enhance their activity [85], disrupting the communication of pathogenic bacteria, and thereby significantly inhibiting the formation of biofilms. The QSI, 3-phenyllactic acid (PLA), produced by *Lactobacillus* sp. is a novel antibacterial substance that inhibits pathogens, particularly fungal infections. The potential of PLA to impede the formation of biofilm is related to its ability to competitively inhibit the pathogen QS system and impact its initial attachment. The inhibitory effects of PLA on QQ have been attributed to its antagonism of the binding of RhlR and PqsR to the QS receptors of *Pseudomonas aeruginosa*. This reflects superior affinity of PLA compared to the homologous ligands of QS receptors [86].

The LuxS/AI-2 bacterial signal system is a universal mechanism that regulates the expression of virulence factors in certain pathogens by influencing their motility, biofilm formation, and attachment. Yan et al. [79] extracted cell-bound BS from *Streptococcus lactis* and *Lactiplantibacillus plantarum*. It was established that these two types of BS affect the expression of biofilm-related genes and inhibited the release of AI-2 in the QS system. The AI-2 activity and virulence factors of an enterohemorrhagic *Escherichia coli* wild-type strain could be significantly reduced via inhibition of the AI-2 signal pathway by *Latilactobacillus sakei*. *Streptococcus pentosus* BS-2 and *Limosilactobacillus fermentum* BM2, which were extracted from milk, exhibited significant quorum quenching activity through their secondary metabolites [87]. Furthermore, the BS-2 and BM2 strains exhibited significant antibiofilm properties against the *Pseudomonas aeruginosa* strain JUPG01, thereby providing guidance for evaluating new candidate LAB strains affecting QQ. Genetic engineering technology can be used to directly metabolize probiotics to produce known substances that inhibit or even degrade the biofilms of pathogens. Two genetically modified *lactobacilli* strains demonstrated significant efficacy in the breakdown of biofilms of pathogenic bacteria through the secretion of a specific enzyme (PelA_h_) derived from the pathogen [88].

### 4.2. Reduction in Surface Adhesion

Postbiotics from LAB exert a protective effect via competitive exclusion, which involves the formation of protective layers to prevent the adhesion and invasion of pathogens. In addition, LAB postbiotics alter the adhesion properties of pathogenic bacteria and impede their colonization on the carrier surface.

BS is a compound that exhibits amphiphilic properties and is produced by a variety of microorganisms [89]. It is classified as a secondary metabolite of microorganisms and encompasses a range of structural groups, such as glycolipids, polysaccharide–lipid complexes, lipoprotein-lipopeptides, and phospholipids. Glycolipid biosurfactants that originate from LAB have the potential to inhibit bacterial adhesion, eliminate biofilms, and mitigate associated infections within the clinical setting and in food production. The dose-dependent inhibitory effects of two types of BS on adhesion and biofilm formation of *Staphylococcus aureus* indicated their potential application in the treatment of biofilm-associated infections caused by this bacterium [72].

BS produced by LAB demonstrated potent antiadhesion and antibiofilm properties by inhibiting bacterial adhesion to surfaces and destroying biofilm through changing the integrity and activity of bacterial cells in the biofilm [75]. As amphiphilic compounds, BS can reduce the interfacial tension between the contact surface and the bacterial surface, thus helping to prevent the formation of biofilms. At the molecular level, BS of LAB have been shown to downregulate the expression of biofilm-related genes. Purified BS extracted from *Lactobacillus* sp. significantly downregulated the expression of glycosyltransferase and fructosyltransferase genes, which play a key role in the initial adhesion of *S. mutans* [90]. *Lactobacillus* strains have been considered as potential postbiotics against *S. mutans* and *C. albicans*. The secondary metabolites of a variety of *Lactobacillus* spp. could maintain stable activity under different physical and chemical conditions [91]. The EPS in postbiotics from LAB can also be attached to the intestinal mucus layer to form an additional protective layer to further prevent the adhesion and colonization of pathogens. Specifically, *Lacticaseibacillus rhamnosus* surface adhesin could compete with oral streptococci, including *S. mutans,* to bind saliva receptors, thereby reducing the adhesion and biofilm production of oral pathogenic bacteria [71].

### 4.3. Antagonistic Effects

*Lactobacillus* spp. are capable of secreting antibiotics, bacteriocins, lactic acid, and other antibacterial substances. These molecules in postbiotics can effectively inhibit the growth of pathogenic bacteria and even result in cell death [76]. Antimicrobial peptides have gained popularity as a suitable alternative to synthetic antibiotics in recent years because of their broad antibacterial spectrum, high antibacterial activity, and unique mechanism [92]. 

Antimicrobial peptides inhibit bacterial adhesion in the initial stage by decreasing bacterial surface charge, hydrophobicity, membrane integrity, and the transcription of adhesion-related genes. Antimicrobial peptides later interact with extracellular DNA, disrupting the three-dimensional structure of mature biofilms, resulting in their dispersion. In addition to antimicrobial peptides produced by LAB, the artificial helical peptide G3 inhibits *S. mutans* biofilm formation by interfering with its biofilm development at various stages [93].

Certain *Lactobacillus* spp. produce bacteriocin, an antibacterial peptide synthesized by ribosomes, which has the advantages of being environmentally friendly and safe. Bacteriocin can also inhibit biofilm formation in a dose-dependent manner; however, it is difficult to eliminate biofilms that have already formed using bacteriocin. The synergistic effect of bacteriocin with other antibacterial agents, the incorporation of nanoparticles (NPs), and application of bioengineering all enhance its antibiofilm activity [94]. Under the influence of bacteriocin and EPS of LAB, the number of living *Pseudomonas aeruginosa* PAO1 cells forming a biofilm in vitro under laboratory conditions decreased significantly [95]. Bacteriocin from *Lacticaseibacillus rhamnosus* XN2 demonstrated antagonistic activity against pathogenic bacteria at the cellular and QS levels, which induced the expression of the luxS gene encoding Al-2 synthetase [96].

In addition to antimicrobial peptides, LAB synthesize a diverse range of small molecular organic acids, hydrogen peroxide, polymeric glycoproteins, EPS, and other compounds during their metabolic processes, which have been demonstrated to possess antifungal properties. The inhibition of fungal growth was significantly influenced by the organic acids generated by LAB [97]. Lactic acid produced by *Latilactobacillus sakei* inhibits *Candida* spp. in general and reduces the formation of *C. albicans* mycelia and early biofilms [98]. Pili affects the initial adhesion of pathogenic bacteria to the carrier surface, and PLA has been shown to downregulate the expression of a pilus gene (*ebpABC*) and a polysaccharide gene (*epaABE*) to inhibit cell movement and EPS production of *Enterococcus faecalis* [99].

Organic acids mostly affect the relative electrical conductivity of the cell membrane of pathogenic bacteria, adversely affecting their plasma membrane integrity and eventually leading to cell death. Various small molecular organic acids in the CFS of LAB are thermostable, but sensitive to pH neutralization. Meanwhile, the pH-neutralized CFS and heat-treated CFS preparation can still limit the formation of pathogenic bacteria biofilm, but to a lesser degree than that of untreated CFS [70]. The CFS of *Pediococcus pentosaceae* 4I1 was identified by gas chromatography–mass spectrometry (GC-MS) to mainly contain organic acid compounds such as amino acids and fatty acids, and effectively inhibits the biofilm formation of foodborne pathogens by severely disrupting the plasma membrane [100]. In the absence of bacteriocins, the three main organic acids produced by *Limosilactobacillus fermentum*, lactic acid, acetic acid, and formic acid, have significant antibacterial effects according to GC-MS analysis [101]. The number of sessile cells of *Salmonella* decreased significantly after treatment with LAB CFS, which proved the potential of LAB as postbiotics to degrade pathogenic biofilms [102].

### 4.4. Regulation of Interspecific Interaction

The interaction among diverse species could interfere with the synthesis of ECM and the distribution of bacterial populations within the biofilm, thereby enabling the indirect control of pathogen biofilms via postbiotics from LAB. In certain environments, LAB can form biofilms composed primarily of EPS, like most bacteria. The majority of LAB in the gastrointestinal tract and in fermented foods exist as biofilms. The ultimate configuration of a biofilm can be influenced by the interplay among distinct bacterial species. It is also a good strategy to control pathogen biofilm by stimulating LAB to form a protective biofilm via their postbiotics. The biofilm phenotype of LAB is advantageous to intestinal homeostasis, tissue colonization resistance, community stability and resilience, and host defense maturity, which regulates the virulence of pathogens and biofilm formation [103].

As a kind of high-molecular-weight polymer, EPSs of LAB have antibiofilm activity and prebiotic properties to stimulate the growth of probiotics [104]. *Lactococcus lactis* efficiently colonizes the human gastrointestinal tract and inhibits the adhesion of pathogenic bacteria to the intestinal mucosa by means of a secretory matrix-based biofilm [105]. The prevention of *Salmonella* infection is attributed to the role played by EPS produced by *Lactobacillus delbrueckii* [106]. Recent research has shown that *Tetracoccus halophilus* biofilms have aggregation and antibiofilm activity against *S. aureus* and *Salmonella typhimurium* [107], and *Lactobacillus* sp. has excellent prospects to control the biofilm and planktonic population of methicillin-resistant *S. aureus*. Confocal laser scanning microscope images revealed that *Lactobacillus* sp. cells within its biofilm were able to capture the biofilm of *L. monocytogenes* in a two-species biofilm system [108].

Cell membrane components, such as lipoteichoic acid, membrane proteins, and lipoproteins, which are included in postbiotics from LAB, can bind to LAB cells or interfere with the recognition between pathogens and host cells [109]. LAB attach to the intestinal epithelial cells via lipoteichoic acid and form a biofilm with other anaerobic species on the mucosal surface of the intestines. This biofilm acts as a barrier that competitively inhibits the adhesion and colonization of potential endogenous and exogenous pathogens to the intestinal epithelial cells. *Streptococcus pentosus* cells bound to lipoprotein exhibited broad-spectrum antiadhesion ability and inhibited the formation of the biofilm matrix structure by affecting the activity and integrity of cells in the biofilm of pathogenic bacteria and decreased the concentration of total ECM [110]. Fibronectin-binding protein A on the membrane of *Weissiella vaginalis* interferes with the invasive pathway of *S. aureus* by competing with it for fibronectin, preventing it from recognizing and binding to the integrin of human breast epithelial cells and inhibiting the formation of its biofilm [111].

Adhesion of intestinal pathogenic bacteria to intestinal cells and subsequent colonization are prerequisites for biofilm formation and virulence. Postbiotics can inhibit the initial colonization of pathogens by enhancing the immune regulation function of the body, especially intestinal immunity. Martorell et al. [112] found that inactivated Bifidobacterium longum retains the ability of antistress injury, and it also reduces acute inflammatory reaction, avoids the destruction of the intestinal barrier, and inhibits the colonization of pathogens by activating pathways related to innate immune function. Other studies have found that the immunomodulatory function of heat-inactivated *Lactobacillus gracilis* TMC0356 could induce macrophages to produce more IL-12 and enhance the phagocytosis of intestinal invasion pathogens [113]. A mouse model experiment found that heat-inactivated Lactobacillus plantarum b240 can effectively protect the host from *S. typhimurium* by stimulating immunoglobulin A secretion [114]. Similarly, heat-inactivated *Lactobacillus parasitum* MCC1849 can induce Ig+ cells in the intestinal tissue to secrete IgA and affect the production of follicular helper T cells in the intestinal collecting lymph nodes, thereby stimulating the host acquired immune response [115].

There are immunomodulatory metabolites in postbiotics of LAB, including LTA, peptidoglycan, and SCFA, which have been shown to affect many immune responses, including inhibition of NF-κB. LTA could interact with Toll-like receptor (TLR) 2 or TLR6 [116], and peptidoglycan or its derived peptides interact with NOD2 [117]. If not destroyed or changed by the inactivation process, these molecular patterns related to microorganisms may also exist in postbiotics. In addition, it has been reported that metabolites such as lactic acid can mediate the immune response through GPR31-dependent dendritic processes in intestinal CX3CR1+ cells [118].

## 5. Discussion

A biofilm comprises a nutrient gradient, and thus the activity of the cells located in the center of the biofilm is the lowest. This gradient provides a material basis for the transfer and latency of pathogenic bacteria in the host body [119]. There is a correlation between the degree of drug resistance of various pathogenic bacteria and the formation of biofilms [120], and it is difficult to eradicate human infections caused by biofilms because their unique structure is resistant to antibiotics [121] and host immune factors [122]. Compared with the mature biofilm, the early biofilm has a more fragile structure, a more robust metabolism, and greater sensitivity to antimicrobial agents. Consequently, it is essential to remove a biofilm at the early stage.

Postbiotics have a wide range of targets, not only in the intestinal tract, but also in the oral cavity, skin, genitourinary tract, or nasopharynx, which can be easily absorbed by the intestinal tract and improve the utilization rate. Therefore, postbiotics can be applied in food, medicine, and other industries. The postbiotics from lactic acid bacteria come from the strains with clear genetic background and biological characteristics and are generally recognized as safe. After safety evaluation, and according to the standard dose when used, the safety can be guaranteed. Postbiotics also adapt to some special groups, such as newborns and sensitive people. For the sake of safety, the addition of exogenous additives such as enzymes and acids and bases should be minimized in the process of postbiotic preparation, and the cleanliness of the production workshop should be ensured at the same time.

From food processing to the transportation and storage of products to the final table, these processes may introduce foodborne pathogens at any time, resulting in a sharp increase in food safety risks. Postbiotics derived from LAB have the potential to serve as food additives, offering both probiotic benefits and biological antiseptic effects. Postbiotics exhibit favorable stability during storage and transportation and are suitable for processing applications. The incorporation of postbiotics will exert a direct influence on the sensory, physical, and chemical attributes of the final product. Therefore, the dose of postbiotics added in the process of food processing needs to be reasonably regulated. Furthermore, it is critical to comprehensively evaluate the interplay between postbiotics and food composition during their utilization. There are some obstacles to the direct use of postbiotics in food processing, such as processing temperature and food surface pH value, which will limit the antibiofilm effect of postbiotics. The standards of many postbiotics products on the market vary, and postbiotics used in the food industry must reach food-grade. The antiseptic effect of postbiotics can be maximized by nano-encapsulation or coating of these compounds in the film. The food processing table and equipment pipeline can easily breed pathogenic biofilms, which are difficult to remove. Postbiotics used as a liquid antibiofilm agent can be sprayed regularly on the surface in contact with raw materials, which is safer than traditional disinfectants without additional risk factors. Although postbiotics have certain advantages compared with antibiotics and chemical preservatives, a single postbiotic has some problems, such as narrow bacteriostatic spectrum and high production cost, which cannot be ignored in its application.

Postbiotics from LAB, particularly *Lactobacillus* spp., have application prospects in clinical medicine as drugs to control biofilm infections. However, there is a lack of dependable and efficient carriers. The upregulation of drug efflux pump gene expression in pathogens is attributed to the presence of biofilms, which, coupled with the physical barrier effect of the biofilms, poses a challenge in directly targeting the pathogenic cells in the biofilm. From the point of view of clinical treatment, postbiotics from LAB are not a perfect drug substitute for antibiofilm action, and there are many cost and technical limitations compared with traditional drugs. First, postbiotics with antibiofilm function are screened from postbiotics from different LAB strains. Because of their complex components, many in vivo and in vitro experiments are needed to confirm their function of eliminating biofilm. The preclinical pharmacodynamics research period is long, and the cost is high. Secondly, compared with drugs, it is difficult to determine the main substances of postbiotics, which influence the setting of the drug dose and pharmacological research. The most important thing is that the postbiotic components of LAB produced in different batches are affected by fermentation, inactivation, concentration, and preparation, and the efficacy of different batches fluctuates greatly, which cannot be standardized for production as commercial drugs at present. The instability of efficacy means that postbiotics can only be used as an adjuvant therapy.

NPs have emerged as a novel antibiofilm functional material that addresses the issue of inadequate targeting of conventional antimicrobials to biofilms [123]. Additionally, NPs serve as effective carriers to deliver natural antibiotics [124]. The increase in QQ microorganisms and the prevention of biofilm fouling can be achieved using NPs, which function by breaking down the QSMs of pathogens and hindering their adhesion and proliferation [125,126]. The combination of bacteriostatic agents and NPs presents a comprehensive enhancement of the therapeutic efficacy of the transported compounds [127]. Chitosan, as a prebiotic, could stimulate the growth of LAB while concurrently impeding the proliferation of pathogenic bacteria. Quantitative real-time PCR analysis confirmed that chitosan extracted from marine biological sources downregulated the expression of genes related to adhesion and virulence of *Serratia marcescens*, such as *fimA*, *fimc*, and *flhd* [128]. Additionally, chitosan downregulated the expression of genes related to the QS system of *Pseudomonas aeruginosa*, specifically *lasI* and *rhlI* [129]. The postbiotic–NP complex functions as a protective complex with the aim of targeting the biochemical composition of ECM to counteract and eradicate biofilms, while simultaneously optimizing the antimicrobial efficacy of these constituents [130]. The combination of *Pediococcus acidilactici* postbiotic with chitosan exhibited a synergistic effect against the biofilm formation of *S. typhimurium* and *L. monocytogenes* during production storage [131]. As a new type of antibiofilm agent, there are still many problems to be solved in postbiotics from LAB. There is an urgent need for excellent carriers to deliver postbiotics in food antisepsis and clinical treatment. The utilization of NP-modified chitosan biopolymer or nano-sized chitosan could serve as a viable delivery method for LAB postbiotics in clinical therapeutic applications. The postbiotic-fortified chitosan coating could be widely used in the field of food preservation and antisepsis, with the aim of prolonging the shelf life of food. It is imperative to conduct a thorough analysis of certain effector molecules through metabolomics and clarify its antibiofilm mechanism. Figure 4 shows the techniques of analysis for the main postbiotic components.

Intestinal flora and their metabolites may interact with each other and cause two-way interaction between the intestinal tract and lung tissue through the blood and lymphatic system [132]. Oral administration of live LAB can increase antibody production, enhance natural killer cell activity, and increase IFN-γ and IL-10 [133,134]. Intake of Lactobacillus and Bifidobacterium can regulate lung immune response through a variety of signaling pathways. This suggests that live LAB are likely to maintain lung health by affecting intestinal microbial metabolism and immune regulation. The microbial metabolic components with immunomodulatory properties in the intestinal tract include indole, nicotinic acid, polyamine, pyruvate, and lactic acid, all of which play an important role in intestinal homeostasis [135]. Some microorganisms in the intestinal tract could use the lactic acid carried by metazoan to produce beneficial SCFAs and butyrate. SCFAs has been confirmed to affect respiratory health. The postbiotics of LAB are rich in a large amount of SCFAs. After entering the human intestinal cavity, they form a local immune response in the intestinal tract, supplying energy to colon cells, especially butyric acid. The excess SCFAs in the intestine, which are not metabolized by the liver, enters the peripheral circulation and bone marrow, affecting the development of immune cells [136]. The communication mechanism of the lung–gut axis may also involve the direct migration of immune cells from the intestine to the respiratory tract through circulation, such as ILC2s, ILC3s, and TH17 cells [137]. Other metabolic components contained in postbiotics from LAB can affect intestinal microbial metabolism and enhance the function of intestinal immune response. At the same time, these immune cells with enhanced immune response levels migrate from the intestines to the lungs, participate in respiratory inflammation, and reduce respiratory infection symptoms. To sum up, oral administration of postbiotics from LAB can improve the structure of intestinal flora and prevent and assist the treatment of respiratory tract infection through the gut–lung axis.

## 6. Conclusions

Although live LAB and their purified metabolites are widely used in the fields of food and medicine, the production and application of postbiotics is still in its infancy. Postbiotics from LAB prevent the formation of biofilm of pathogens in the early stage and still eliminate most of the mature biofilm. From the point of view of the biofilm cycle, the pathogen in the planktonic state is the best stage for postbiotic mediation. The antibiofilm mechanism of postbiotics from LAB in vitro mainly affects the surface adhesion, self-aggregation, virulence, and QS of pathogens. The antibiofilm mechanism in the host involves beneficial regulation of the microbiome, enhancement of epithelial barrier function, regulation of immune response, and indirect regulation of host microbiome by QS and QQ molecules that may exist in postbiotics. Oral postbiotics can improve the structure of intestinal flora and assist the treatment of biofilm infection. Some difficulties have been found in the practical application of postbiotics. Postbiotics are relatively stable and safe, but they require special carriers to better exert their antibiofilm effect and a large number of clinical trials are still needed to verify postbiotic safety. For industrial mass production, heat-killing techniques and time, as well as the addition of exogenous materials, will have an impact on the characteristics of postbiotics. Therefore, careful consideration of carrier selection and process parameters is crucial to ensure optimal outcomes when utilizing postbiotics as a strategy to control biofilm.

In summary, continuously improving the definition of postbiotics and formulating process specifications and technical standards is a crucial objective in the field of probiotics. The realization of quality control and supervision of postbiotics is dependent on the definition of characteristic components of postbiotics’ health effects and the establishment of qualitative and quantitative analytical methods. With the continuous development of clinical trials of postbiotics in the treatment of dental caries and respiratory tract infections, how to further optimize the preparation process and reduce the composition differences between different batches has become a key research direction. Compound fermentation metabolites and inactivated LAB may broaden the bacteriostatic spectrum, and multi-strain fermentation of postbiotics can also be considered in the future.

## Figures and Tables

**Figure 1 microorganisms-12-00704-f001:**
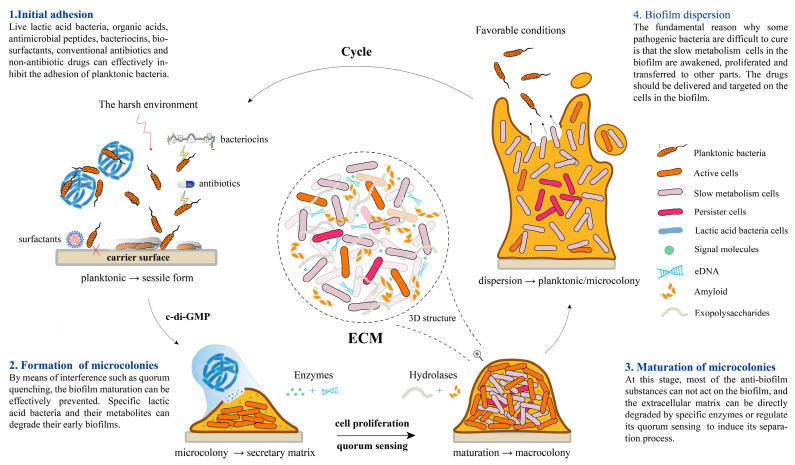
The life cycle of bacterial biofilm and control methods in each stage. Pathogenic bacteria resist environmental stresses by aggregating to form biofilms, then will separate and spread under favorable conditions. eDNA, extracellular DNA; ECM, extracellular matrix.

**Figure 2 microorganisms-12-00704-f002:**
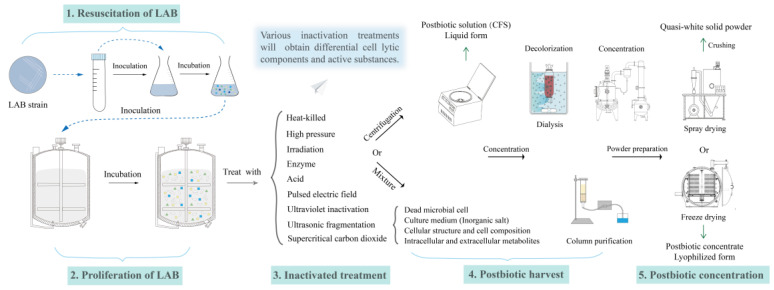
The universal processing methods of postbiotic preparation produced by LAB in liquid or solid powder forms. Avoiding introducing exogenous substances, physical inactivation is the optimal choice, especially under high pressure. CFS, cell-free supernatant.

**Figure 3 microorganisms-12-00704-f003:**
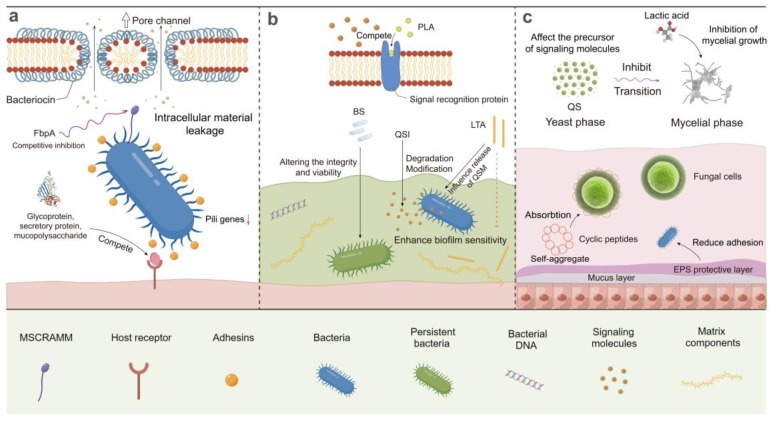
The main mechanisms of controlling film-forming pathogens in biofilms and host environment via postbiotics from LAB. (**a**): Directly kill planktonic bacteria and prevent them from recognizing and adhering to the host. (**b**): Inhibit maturation by affecting signal transmission and biofilm stability. (**c**): The organic acid and small molecule peptides in postbiotics prevent the colonization of pathogenic bacteria in the intestinal tract and inhibit the transformation of fungi to mycelial phase. QS, quorum sensing; QSI, quorum sensing inhibitor; QSM, quorum sensing molecule; FbpA, fibronectin-binding protein; EPS, extracellular polysaccharide; MSCRAMM, microbial surface components recognizing adhesive matrix molecules, such as fibronectin-binding proteins and biofilm-associated protein; BS, biosurfactant; LTA, lipoteichoic acid; PLA, 3-phenyllactic acid.

**Figure 4 microorganisms-12-00704-f004:**
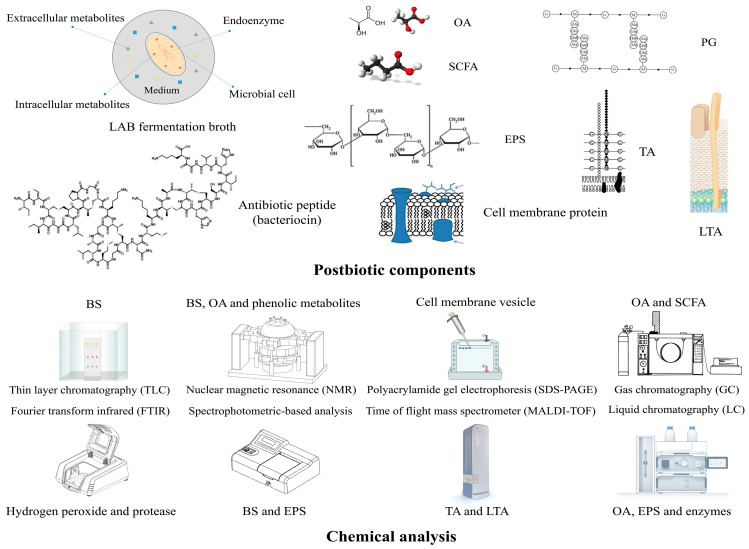
Main postbiotic components of LAB and common analytical techniques. OA, organic acid; SCFA, short-chain fatty acid; PG, peptidoglycan; TA, teichoic acid; LTA, lipoteichoic acid.

**Table 1 microorganisms-12-00704-t001:** Antibiofilm mechanism and application of LAB postbiotics.

Name of LAB	Postbiotic form	Functional Components	Mechanism of Antibiofilm	Application Potential	Reference
*Lb. acidophilus*	CFS	Exopolysaccharides and biosurfactants	Inhibition of initial adhesion to the biological surface	Controlling or preventing ESBL colonization or infection	[69]
*Lactobacillus* spp.	CFS in neutralized and heat-treated form	Organic acids (lactic, formic, and acetic acids) and bacteriocins	Inhibition of initial adhesion to the biological surface/promotion of the dispersion of biofilm	Bio-control agents used to prevent infections	[70]
*Lactobacillus* spp.	Heat-killed cells/CFS	hydrogen peroxide, bacteriocins and biosurfactants	Competition for the specific salivary receptors/displacement of biofilm via high co-aggregation ability	Products for oral hygiene	[71]
*Lb. acidophilus* LA5 *Lb. casei* 431	CFS	Exopolysaccharide and biosurfactants	Inhibition of adhesion to the biological surface	Biofilm removal compounds to control the foodborne pathogens	[72]
*Lb. acidophilus* La14 150B *Lb. plantarum* B411 *Lb. rhamnosus* ATCC 53103	CFS	Organic acids (lactic and acetic acids) and bacteriocins	downregulation of *prfA* gene involved in biofilm formation	Food-grade sanitizers	[55]
*Lb. rhamnosus*	cell-free extracts (solution)	oleic and myristic acid	Downregulation of virulence gene/inhibition of yeast-to-hyphae transition	Therapeutic agents to treat *Candida* infections	[73]
*Enterococcus* sp. CM9 and CM18	CFS	bacteriocins	Competition for adhesion site	Control of food contamination	[60]
*P. acidilactici* 27167 *Lb. plantarum* 27172	cell-free extracts (solution)	biosurfactants	Reduction in expression levels of biofilm-related genes/interference with the release of AI-2	Therapeutic agents to treat *S. aureus* infection	[74]
*L. mesenteroides* J.27	CFS (lyophilized)	Organic acids (lactic, acetic, and citric acid)	downregulation of the mRNA expression of virulence-related genes	Green preservative in seafood processing	[63]
*Lb. rhamnosus*	CFS (lyophilized)	Glycolipid biosurfactants	Inhibition of initial adhesion to surfaces/altering the integrity and viability of biofilm cells	Green antibiofilm agents	[75]
*Lactobacillus* spp.	CFS	biosurfactants, lactic acid, and exopolysaccharides	Inhibition of initial adhesion to surfaces/induction of pore formation on the bacterial cell surface/suppression in short-chain AHL production	The prevention and treatment of orthopedic infection	[76]
*Lb. fermentum KT998657*	CFS in neutralized/cell-free extracts (solution)	Exopolysaccharides and bacteriocins	Reduction in quorum sensing signals needed for biofilm formation/matrix modification/restriction on cell assembly and attachment	Prophylactic agents for medical devices	[77]
*Lb. plantarum* FT 12 *Lb. brevis* FT 6	CFS	Organic acids and bacteriocins	Interfere with quorum sensing/high co-aggregation ability with pathogens	Supportive oral health treatment	[61]

## Data Availability

Not applicable.

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
