# Peer review of "Elimination of Pathogen Biofilms via Postbiotics from Lactic Acid Bacteria: A Promising Method in Food and Biomedicine"

_microorganisms, 2024, doi:10.3390/microorganisms12040704_

Round 1
Reviewer 1 Report
Comments and Suggestions for Authors
The authors conducted study titled "Elimination of Pathogen Biofilms via Postbiotics from Lactic Acid Bacteria: A Promising Method in Food and Biomedicine".
In attachment are some comments in order to improve this Manuscript.
The paper presents an overview about the potential application of postbiotics derived from LAB to control microbial biofilms and their mode of action. The overall quality of the review is good and authors wrote a significant topic that can be relevant for the microbiology community. However, I provide some comments that should be addressed before this manuscript could be published. The following suggestions are presented:
Specific points
- Line 165: Briefly describe the Method for isolation of postbiotic from LAB and specify which method is used?
- The authors should discuss in more depth about the safety aspects, the obstacles, and future perspectives of using postbiotics in the food industry.
- In conclusion add a few sentences about innovative data from this review that will open up new insights for food applications of postbiotics prepared from LAB.
- Try to conclude with a general statement of the most relevant part of this study.

Reviewer 2 Report
Comments and Suggestions for Authors
This manuscript is very interesting and summarizes the biofilm life cycle as well as emphasizes the importance of quorum sensing in the maturation and dispersion stages. However, here are some considerations for improving the manuscript:
- Reformulate the abstract to clearly contain the type of study, objective and conclusion.
- Add the study aim at the end of the manuscript introduction.
- Add "spp." without italics in table 1.
- Add study limitations and future perspectives to the manuscript discussion.
- The manuscript conclusion must be modified to be clearer and more synthetic.
- All content of the manuscript conclusion must be transferred to the discussion: "A biofilm comprises a nutrient gradient, and thus the activity of the cells located in the center of the biofilm is the lowest. This gradient provides a material basis for the transfer and latency of pathogenic bacteria in the host body [120]. There is a correlation between the degree of drug resistance of various pathogenic bacteria and the formation of biofilms [121], and it is difficult to eradicate human infections caused by biofilms because their unique structure is resistant to antibiotics [122] and host immune factors [ 123]. Compared with the mature biofilm, the early biofilm has a more fragile structure, a more robust metabolism, and greater sensitivity to antimicrobial agents. Consequently, it is essential to remove a biofilm at the early stage. The postbiotics from LAB comprise a mixture of inactivated bacterial cells, bacterial components, and metabolites. It is imperative to conduct a thorough analysis of the constituents and assess the safety of postbiotic formulations. Fig. 4 shows the means of analysis for main postbiotic components. Improving the definition and processing standards of postbiotics is a crucial objective in the field of probiotics. The realization of quality control and supervision of postbiotics is dependent on the definition of characteristic components of postbiotics' health effects and the establishment of qualitative and quantitative. With the introduction of the concept of postbiotics, how to create and make use of stable LAB strains has become a key research direction. The analysis of the effective components of postbiotics and the efficient preparation of stable postbiotic preparations is a pressing issue given the numerous metabolites of LAB.".
- The manuscript conclusion must not contain references.
Reviewer 3 Report
Comments and Suggestions for Authors
Overall, this review is nicely structured and could be further benefited with some minor comments.
1. Are there some limitation with the use of probiotics? Please include a separate section.
2. Please enlist the commercially available or under clinical trial probiotics with anti-biofilm activity.
3. Please discuss the lung-gut axis and role of probiotics in respiratory infections, especially with focus on the applicability of probiotics as medicine for respiratory infections. This is important. Here is one reference. https://pubmed.ncbi.nlm.nih.gov/33425362/
4. Authors should discuss about the immune response of probiotics. Please justify and discuss this important point.
Round 2
Reviewer 2 Report
Comments and Suggestions for Authors
The authors made all suggested adjustments.